# A Proof-of-Concept IoT System for Remote Healthcare Based on Interoperability Standards

**DOI:** 10.3390/s22041646

**Published:** 2022-02-19

**Authors:** Lenin-Guillermo Lemus-Zúñiga, Juan M. Félix, Alvaro Fides-Valero, José-Vte. Benlloch-Dualde, Antonio Martinez-Millana

**Affiliations:** 1Instituto Universitario de Investigación de Aplicaciones de las Tecnologías de la Información y de las Comunicaciones Avanzadas (ITACA), Universitat Politècnica de València, Camino de Vera s/n, 46022 Valencia, Spain; juafeis@inf.upv.es (J.M.F.); alfiva@upvnet.upv.es (A.F.-V.); anmarmil@itaca.upv.es (A.M.-M.); 2Computer Engineering Department, ETSINF, Universitat Politècnica de València, Camino de Vera s/n, 46022 Valencia, Spain; jbenlloc@disca.upv.es

**Keywords:** IoT, sensors, interoperability, ehealth, e-health, health data, FHIR, universAAL

## Abstract

The Internet of Things paradigm in healthcare has boosted the design of new solutions for the promotion of healthy lifestyles and the remote care. Thanks to the effort of academia and industry, there is a wide variety of platforms, systems and commercial products enabling the real-time information exchange of environmental data and people’s health status. However, one of the problems of these type of prototypes and solutions is the lack of interoperability and the compromised scalability in large scenarios, which limits its potential to be deployed in real cases of application. In this paper, we propose a health monitoring system based on the integration of rapid prototyping hardware and interoperable software to build system capable of transmitting biomedical data to healthcare professionals. The proposed system involves Internet of Things technologies and interoperablility standards for health information exchange such as the Fast Healthcare Interoperability Resources and a reference framework architecture for Ambient Assisted Living UniversAAL.

## 1. Introduction

In contrast to reactive care, preventive health care is based on the promotion of health and continuous control of health as an anticipated way for preventing diseases [1]. Beyond epidemiology, the paradigm of disease prevention has improved dramatically in the last decade, leading to an increase in interest in the management of data related to patients’ health from a molecular level to a public health dimension [2]. Despite this growth, little of the information collected by devices outside the healthcare system is used in diagnosis, treatment and follow-up. This hampers the potential benefits of fields such as machine learning or big data analytics [3,4]. A recent review on the reference architectures and platforms for European Smart and Healthy Living [5] analyzed the features of the Ambient Assisted Living (AAL), which included smart home monitoring, robotics, wearable and assistants. The authors of this review concluded that the poor generalization of data collection and exchange standards became one of the main problems of interoperability in this field; moreover these systems should seek the connection to healhcare professionals, a main barrier which explains why many of the clinical data are not being used [6,7]. The IoT paradigm involves electronic devices, usually self-powered for several years, that can operate between themselves without human intervention and at a high performance [8,9]. The new services in the information society, which involve mobile devices and social media, are being adapted for the promotion of health and to support a healthy and active ageing, both in European countries and in the United States [5,10]. However, the current literature is focusing in the back-end at system level rather than the actual end-to-end approach. As an example, architectures have been proposed for implementing interoperable services using ontologies and semantic reasoning [3], and focusing in the quality of data collected by medical devices [11]. The study presented in this paper addresses the design and implementation of an interoperable system that allows the collection, monitoring and transformation of biometric data based on rapid-prototyping technologies and sensors for later use in healthcare environments. The main objective is to design and deploy a proof-of-concept of an interoperable system based on IoT elements for the telemonitoring of elderly people in home environments with direct connection to healthcare professionals. The focus on IoT has the objective of using computing elements that allow working with less power consumption than current systems based on computers and to implement a system that allows machine-to-machine communications. To achieve this main objective, three secondary objectives were defined: (1) to integrate a set of biometric sensors capable of collecting patient measurements and transforming them into useful data; (2) to develop a middleware capable of bridging the sensors and a server in the cloud based on IoT computing elements; and (3) to guarantee the interoperability of the system through the implementation of standards capable of allowing the integration of the system with external platforms with other systems compliant with the same standard.

## 2. Related Work

One of the biggest drawbacks of existing solutions focused on IoT and Health technologies is the lack of standards for the management of data from different heterogeneous devices. For example, the *Personal Health Device Observation Consumer* [12] consists of a monitoring system that collects data from a pulse oximeter sensor through a Raspberry Pi and transforms the data to the FHIR standard through the use of JSON messages. One of the advantages of using the Raspberry Pi as an adapter is its power to increase the number of messages that can be sent to the server. Once the data is transformed, RFC 7252 Constrained Application Protocol (CoAP) messages are sent at regular time intervals thanks to the use of the POST method to create, and the PUT method to update. Subsequently, a middleware that acts as a bridge between the application and the FHIR server is instantiated. It also functions as a monitoring system for the different devices since it includes the functions of registering and monitoring the system. The *eHealthNet monitoring* [13] solution enables acquiring data from different sensors for the treatment of cardiovascular diseases. Electrocardiogram (ECG) sensors are used to collect data on the pressure with which the heart performs beats. This data is collected by a middleware via Bluetooth and stored in an internal SQLite database, increasing the persistence of the system. The patient, in turn, can also enter data manually if there is a case of bad reading. Once transformed to the standard, the data is sent to an FHIR server. The doctor can consult the patient’s data once the data has been uploaded to the server. Franz et al. proposed a smartphone-based system for the exchange of data based on the FHIR standard [14]. The sensors send information via Bluetooth to a Java application on an Android terminal. The middleware decodes the information in a readable format and the results are transferred to a centre for health information exchange services (Telehealth Service Center). The lack of a local database that allows the handling of the data means the persistence of these is reduced, and there may be lost data packages in the medium and long term. Moreover, the authors make note of the low battery life of smartphones due to high data traffic. Virtualized cloud-based systems have also been proposed to maintain the integrity and full availability of such systems for critical operations, such as emergency care [15]. More recently, the paradigm of IoT for medical healthcare applications was proposed in [16], which differs from the general IoT by its resource-restricted context and heterogeneous technological set up. New algorithms for efficient power supply systems and better throughput are constantly being deployed, helping to achieve the goal of more reliable systems. In the area of commercial solutions, *My Fit* is the official application for Xiaomi fitness devices. Despite not being a pure IoT device, it has an Android application that allows the monitoring of heterogeneous devices in a simple and unified way. One of the disadvantages of this type of application compared to other monitoring applications is that they use their close standard of data communication, preventing them from being interoperable between non-brand devices. The application allows the insertion of data manually, being able to keep track of your weight without the need to have a conventional commercial scale.

IoT healthcare applications include a middleware that acts as an intermediary between the sensors and the data-processors or services in which the data will be eventually stored. Usually, this middleware performs a double function. On the one hand, it acts as a monitoring system, allowing the user to supervise data by themself. On the other hand, it is used to gateway the information received by the sensors for the system backend, and in some cases this is based on a healthcare information exchange standard. Table 1 presents a summary these solutions.

Current existing solutions in the literature have not coped with the problem of scalability, data persistence and portability and, importantly, the proposed solutions do not incorporate all the technologies from end-to-end: sensors, data collectors, gateways and platform. The objective of the present work is to design a plug-and-play system, capable of integrating low-cost sensing technologies and inexpensive computation resources compliant with international standards.

## 3. Materials and Methods

This section contains a high-level overview of the architecture of the proposed solution, defining the goals, use cases and components that have been selected for implementation and validation. The elements of innovation in the proposed solution are components that have been successfully implemented in state-of-the-art solutions, but that have not been assembled together and tested as a system yet. Before defining the technicalities of the system, this section presents its intended users and functionalities. Thereafter, the section introduces the elements for innovation, such as low-cost healthcare sensors, the Ambient Assisted Living platform and the standards for health information exchange.

### 3.1. Intended Users and Design

The purpose of this study is to create a monitoring system for dependent or elderly people who need constant care. Moving beyond the current state of the art, the system must include a component for healthcare supervision and must feature a standard of health data exchange, capable of interacting with a reference architecture for AAL. The proposed system is presented using the Vision of Philippe Kruchten’s “4 + 1” model [17], which allows different stakeholders to find what they need in the software architecture more quickly. Considered stakeholders are:User—user to whom the system is intended. The user interacts with the available graphical user interfaces to initiate and monitor their activity in the application. The user is capable of executing easy tasks and performing self-monitoring routines;Health professional—user who acts as a mediator in the management of data between the patient and the application server. They can observe and monitor the data of the users, as well as register, modify and eliminate sensors from the system.Administrator—in charge of the global management of users and devices. The administrator can register new entities into the system, modify the data of each user/device, or cancel them from the system;External agent—external user/entity to the system that can use part of its functionality, making connections through the use of the provided interfaces;Interoperability services—a link between the proposed system and legacy/existing systems which provide another type of functions and store complementary data.

The use cases considered defining the logical view, the process view, the development view and the physical view of the system are described in Table 2.

### 3.2. Hardware

The resource-constrained nature of development boards combined with the diversity of hardware architectures and operating systems is common to find in IoT environments [18]. Thus, to overcome this challenge in our proof of concept, we chose friendly and widely used development kits designed for developers and researchers, such as: (i) a Raspberry Pi 3B (https://www.raspberrypi.org/products/raspberry-pi-3-model-b, accessed on 1 November 2021) mini-computer equipped with Microsoft Windows IoT to acts as an IoT gateway; (ii) Arduino UNO (https://store.arduino.cc/arduino-uno-rev3, accessed on 1 November 2021) to define the communication protocol between the IoT sensors and the mini-computer, through an internal function that was programmed to acts as a loop to send the data measured as messages encoded in JavaScript Object Notation (JSON) format; (iii) the e-Health Sensor Shield (https://www.cooking-hacks.com/ehealth-sensors-complete-kit-biometric-medical-arduino-raspberry-pi, accessed on 1 November 2021) to connect different medical and biometric sensors, such as oxygen in blood, body temperature, among others; (iv) a Body temperature Sensor (https://www.cooking-hacks.com/pulse-and-oxygen-in-blood-sensor-spo2-ehealth-medical, accessed on 1 November 2021) capable of measuring corporal temperatures between 0 and 500°. It is connected to the e-Health shield jack port; (v) a Pulse and Oxygen in Blood Sensor (SPO2) (https://www.cooking-hacks.com/body-temperature-sensor-ehealth-medical, accessed on 1 November 2021) that allows the percentage of oxygen saturation in blood to be determined using non-invasive photoelectric pulses. It must be placed on a part of the body that has good blood flow, such as the fingers. A diagram of these components can be seen in Figure 1. These sensors have been already tested previously in other eHealth applications [19].

### 3.3. Interoperability and Standardisation

IoT devices specialized in health care have increased by 46 million units in 2015, and by 161 million in 2019 [20]. In addition to the countless different vendors of devices present in the market, this growth makes it more difficult to reach an optimal IoT architecture that guarantees that the sent data from one device could be interpreted correctly in another place.

Currently, there are multiple scenario-driven architectures and protocols for the IoT, especially focused on smart cities, smart grids, industry and smart cars. The OSI reference model of communications defined seven levels (from physical to application) in which the information is formatted and wrapped into different ways to accomplish a goal, such as to communicate two nodes successfully. Architectures in the IoT are distributed, which means that, usually, sensors have no processing or storage capacity on the data they acquire from the environment [21]. Data should be moved (transmitted) to another computer through a communication channel in a specific way. Many of the traditional communication protocols of OSI model are still valid in the IoT and their future improvements will be key. This is the case of local network connections via Ethernet or WiFi. For example, these two options are foreseen by *Vodafone* and *IBM* in their connected cities (https://www.ibm.com/products/software#addendum-technologies, accessed on 1 November 2021), where the connection speeds that will allow the next protocols, such as 5G, to be the basis of the IoT’s long-range connectivity. But there are also new protocols that have been designed taking into consideration the IoT and the communication of objects in a close range. An example is NFC or Bluetooth 4.0, which has the surname of LE ’Low Energy’ because it is intended to be implemented in systems with reduced batteries such as quantifying wristbands and sensors. In this context, oneM2M standard proposes a simple three-layer model comprising applications, services and networks [22]. So, several apps, databases and systems that can share information complying with the security and privacy requirements are desired to be implemented in healthcare entities as part of their integration of information efforts. Here, we describe briefly two emerged initiatives to approach healthcare interoperability that were used in the system.

#### 3.3.1. Fast Healthcare Interoperability Resources (FHIR) Standard

The International Healthcare Technology Standards Developing Organization (IHTSDO) promotes the preferred terminology, standards and protocols to be independent of vendor platforms [23], e.g., Health Level Seven (HL7) is one of the formats used for information interchange among its different version. FHIR was developed by taking the best features of other HL7 standards and Clinical Document Architecture (CDA), FHIR combines them with widely used web standards HTTP, RESTful, XML and JSON [24].

The REST API of FHIR includes a “unifier”, which transforms the input data into generic units and an “analyzer” which functions as the engine for the advanced analysis of input data [13]. It is composed of a modular architecture, giving the possibility of adding new features, adding new modules [25].

#### 3.3.2. Ambient Assisted Living (UniversAAL)

The universAAL platform (https://www.universaal.info/, accessed on 1 November 2021) is one of the biggest solutions focused on the use of information and communication to improve the quality of life of people with disabilities and the elderly [26,27]. The platform enables developers, vendors and researchers to provide better solutions with a feasible, economic technology and integrated into a large platform to help and assist people with disabilities or the elderly. In contrast to other existing platforms, the universAAL platform aims to connect different heterogeneous devices in a single unified network focused on ambient assisted living scenarios. One of the advantages of universAAL is that it is compatible with a wide range of IoT protocols and that represents information using ontologies [28]. Ontologies are a way of representing the information of everyday life, and a way of being intelligible to computers; it is a network of concepts linked by properties [29,30,31].

The universAAL platform is a universal infrastructure system for AAL software and service businesses, serving and boosting the market for AAL applications and solutions. universAAL provides an open-source runtime, ancillary software and documentation for creating AAL applications and a web store for selling both applications and services. Based on open, vendor-neutral technology standards, the universAAL platform provides an industry-oriented solution that enables creating end-to-end systems providing a dedicated Runtime Platform, storage services, an integrated development environment, integration of Android-OSGi applications, integration of web-services and a unique reference model.

The universAAL Runtime Platform provides an environment for effortlessly deploying AAL services at home. It represents the state of the art in software platform technologies to facilitate the development of AAL solutions. Value is easily understood by system architects, project managers and development engineers who find a comprehensive set of features for interoperability, reasoning, device connectivity, data exchange, accessibility of the user interface, deployment in different nodes, security, scalability and other minor characteristics that make it an attractive product.

Another advantage of using universAAL is the Reference Architecture, which sets and defines terms of the domain, the basic architectural choices and the foundation of the basic design principles. The universAAL reference Architecture provides a conceptual framework for developing AAL services. A software architect should revise the conceptual framework to find out if the proposal fits the requirements of the system under development. If the basic requirements are satisfied, that is, the architecture proposal fits the idea of the desired system, then the designer will feel comfortable in adopting the architectural proposal of the Reference Architecture. Adopting the reference architecture will allow more interoperable services to be had in the future through assuming design decisions already considered by a community of experts in the AAL domain. Using AAL Domain ontologies, developers can find predefined domain ontologies, well-structured and validated in the use of different services. Using universAAL ontologies will ensure direct interoperability of new services with existing previous services.

### 3.4. Software

Regarding the software part, below we describe how the data flows between the IoT sensors through the universAAL server up to the final user devices. The process view developed to fulfil the system requirements is described in Appendix A, whereas the logical view, the physical view and the development view are shown in this section and subsections. The application was designed to run on a wide variety of devices, this is possible by the use of the Universal Windows Platform (UWP) (https://docs.microsoft.com/en-us/windows/uwp/get-started/universal-application-platform-guide, accessed on 1 November 2021). Using Visual Studio 2019 as an Integrated Development Environment (IDE) the application was modelled, developed and tested. It was written in C#. Finally, it was deployed on the Raspberry Pi. The carried-out work has been divided into three blocks: the first block establishes a communication protocol between the IoT sensors and the mini-computer (IoT gateway); in the second block the data is collected, transformed and stored locally; and finally, in the third block, the data is validated by the FHIR format and send it through REST API to the universAAL server. Figure 2 shows a diagram summarising the architecture; below, these modules are described in detail.

#### 3.4.1. Communication Protocol

The Raspberry Pi and the Arduino are connected with a USB cable and the protocol establishes that the Raspberry Pi requests an update to the Arduino, which thereafter sends the sensor data in a data packet. The packet follows a key-value mapping defined by JSON format, with the name of the sensor and their measured values. The Arduino shield can connect several IoT sensors, allowing to send data points one-by-one or multiplexed in an array. The code in the Arduino side is written in C++ using its Desktop IDE (https://www.arduino.cc/en/Main/Software, accessed on 1 November 2021). The code consists of two fundamental functions: on one hand, the setup to initialize the required components to connect the Arduino with the e-Health sensor shield, using the e-Health libraries (https://www.cooking-hacks.com/media/cooking/images/documentation/e_health/eHealth_arduino.zip, accessed on 1 November 2021) that provide several methods to communicate with the IoT sensors; on the other hand, the loop that is constantly looking for data to write in a buffer.

The FHIR transformator protocol is based on the libraries provided by HL7 (https://www.hl7.org/fhir/http.html, accessed on 1 November 2021) for data verification and transformation. The protocol first validates that the sensor class meets all the mandatory fields as defined by the standard. To this end, the system implements an echo service which returns a JSON-composed message with the sensor information. Once the sensor entity is validated, the protocol performs the same operation for the patient and the doctor entity. Once the basic transactors of the FHIR are validated, the protocol constructs the message according to the standard specifications with the provided Application Program Interface and completes the parameters corresponding to the sensor measurement. Once the message is built, it can be serialized for internal storage or cast to the subscribers.

#### 3.4.2. Persistence

The persistence layer of the system is implemented over the Entity Framework (https://docs.microsoft.com/ef/, accessed on 1 November 2021) since it is a tool that allows handling objects simple and effectively in addition to a set of methods to make queries to the database. Thus, the application can store in real-time the data in the local database using the Entity Framework. The database schema designed for our system is shown in Figure 3.

## 4. Results

The implemented system was deployed using Microsoft .NET technologies, implementing a local database (SQLite) for data persistence in the middleware level (beyond cloud replication in the universAAL system). The use of SQLite library was selected over other relational systems such as MySQL due to the advantages it offers in its local use, focusing on disk read–write operations, increasing the portability of the system.

The system communicates with the REST API of universAAL. Defining the communication of the system and how it interacts with these external systems is one of the major concerns of architecture. Moreover, another important result is how the system and sensors communicate. Due to the behaviour of the system, it can be established that the sensors act as an active system, continuously sending information (PUSH); or by requesting the information when necessary from the system (PULL), that is, passively [32].

### 4.1. System Logic Layer

The main objective of the logic layer is to define the components that the system has and define the interfaces through which the different components communicate with each other. One of the goals of this layer is to isolate the components that have a high probability of being modified from the rest of the system. By defining these interfaces for these components and concealing the internal implementation of the rest of the system, the impact is minimized. Figure 4 describes visually the system architecture and Table 3 describes each of the main components.

The system was developed in Universal Windows Platform (UWP), a platform that has an intuitive interface and is easy to deploy in the Raspberry Pi. UWP is an API created by Microsoft and included in Windows 10 to allow developers to create and deploy applications on a wide variety of devices. This platform supports programming languages such as C++, VB.NET or XAML and C#.

Visual Studio 2017 was the IDE chosen to develop the main project, together with Visual Studio 2015 to carry out the modelling project (some features are not implemented in 2017 edition). The performance tests have also been done with the new version of Visual Studio 2019, but it lacks compatibility with some of the necessary libraries. In the persistence layer of the system, we used Entity Framework since it is a tool that provides a simple and effective set of methods to make queries to the database, allowing developers to handle objects instead of using SQL language. To facilitate calls to the REST API, RestSharp (http://restsharp.org/, accessed on 1 November 2021) library was used, some libraries that aim to reduce the amount of code and simplify POST methods. UWP does not have tools for the design of monitoring graphics. We can solve this problem thanks to the Syncfusion libraries (https://www.syncfusion.com/, accessed on 1 November 2021). This set of libraries provides a series of controls and frameworks for a wide range of programming languages. In our case, we will mainly use the controls to make graphs for both the temperature sensor data and the pulse oximeter data.

### 4.2. Graphical User Interfaces

The graphical user interfaces were designed to provide a simple way to be used in a health care environment (user/patient, doctor and administrator).

The Patient role was dedicated to a person who will be encouraged to perform daily monitoring sessions to keep track of their overall status. The user can perform two operations: to manually introduce data or sensor monitoring in real-time. The user also has the possibility of accessing historical data for a preselected time range. An example is shown in Figure 5a—the data range of body temperature (*Y*-axis) for 1 month until 14 December 2018 (*x*-axis).

The Doctor role was focused on an individual authorized by a health care entity to provide medical services. Specifically, in the proposed system, this role is responsible to follow-up with multiple patients and also to manage their sensors. Figure 5b shows an example of the data collected by the pulsioximeter sensor (SPO2) over 10 months, until the end of May 2019.

Finally, the Administrator role is in charge of managing system users (patients/doctors) and sensors. Available sensors are classified and listed by their functionality, and the administrator can assign one or more to the patient. Whenever a sensor is selected, the whole data is displayed to the doctor.

To ensure the success of the implementation of the application, we have focused on the use of acceptance tests, verifying that the system works properly. These tests favoured an early identification of possible errors within the functional requirements.

### 4.3. Validation

Using .NET FHIR libraries, the objects in the system database (patients, doctors, sensors, etc.) are converted to FHIR resources. The official package for FHIR validation was used to validate the application packets (Library:org.hl7.fhir.validator.jar, accessed on 1 November 2021). In FHIR there are different resources, e.g., for the validation of an IoT sensor the “Device” resource of FHIR was used, because it is an article that is used in the provision of medical care. The result of the evaluation for an IoT sensor can be seen in Figure 6.

Once the correct validation of the system components has been verified, the universAAL service starts. It requires to have a universAAL server working, our prototype uses Apache Karaf (https://karaf.apache.org/, accessed on 1 November 2021) that is an Open Service Gateway Initiative (OSGI) container (https://www.osgi.org/, accessed on 1 November 2021) where applications and components can be deployed; it is controlled by a console terminal as can be seen in Figure 7.

The publisher is responsible for sending data to universAAL, where different external agents are waiting to process new incoming data. All the POST methods that have been sent from the application to univesAAL server are registered on the event manager. So, once the service is deployed in our system, all the events will be received by the console. To verify the correct operation with the server, universAAL has an event manager that keeps a log of all the POST calls that have been sent from the application. So, once the service is deployed from our system, all the events will be received by the external integration service provider.

## 5. Discussion

This paper describes the design and implementation of an interoperable proof-of-concept system including different IoT devices at different layers and the integration with a reference architecture for Ambient Assisted Living. The system has been developed to contribute for improving the health of dependent or elderly patients who need continuous remote monitoring using low-cost technologies. The proposed system provides connectivity and device management, acquiring biomedical signals from an Arduino-based set of sensors that can collect medical data from patients and transform/visualize them for later use in data analysis. The hardware of the system has been designed to accommodate sensors through a proprietary Arduino-based shield and be linked to a Raspberry Pi. The Raspberry Pi provides all the sufficient hardware resources to interface the sensors straight away (no need to use the Arduino), but we have proposed to distribute the computation capability, emulating a distributed agents architecture. Therefore the Arduino node can implement signal-processing tasks and only submit to the Raspberry Pi meaningful information (not only raw data) through wireless interfaces.

The system stands as a scalable link between the IoT sensors and a REST API service with the use of POST calls to publish the data, able to maintain the persistence implemented in a local SQLite database. One of the major contributions of the proposed system is that it features the FHIR standard for health data exchange so that can be integrated into compliant medical and scientific applications. Another significant strength is the semantic interoperability with the universAAL system. The design process was based on architectures and features of existing prototypes [12,13,14], discovering the limitations and drawbacks that these solutions had. Besides other systems, this proposal provides an end-to-end solution, including sensors and applications to be used by patients. To the best of our knowledge, state-of-the-art systems focus on the back-end processing of data and do not describe the last-mile implementation of the system [3,11,15].

The proposed system improves the characteristics of [12] by adding elements to ensure the scalability (low cost implementation with Arduino+ sensors) and to gain persistence with the use of the local data base in the RaspberryPi. Moreover, it reduces the usability barriers to elderly users because the system is autonomous and does not need to be operated with a smartphone (as the systems proposed in [13,14] do). By the use of a big touchscreen, with a client application specifically designed to check biometric signals and to communicate with the system, the prototype can be installed and used by elderly users with mild cognitive impairments.

Among the major challenges of this work, we highlight the following. The Entity Framework Core libraries do not support Windows Universal projects. This problem prevented, a priori, the use of Entity Framework Core in the system. For this reason, we have created three projects within the solution, in which the main project is placed within a shared library and a separate .NET Core console application is used to execute the commands. Another challenge has been to deploy the UWP application into the Raspberry Pi 3. Visual Studio has different ways to debug applications but if external libraries are used, the built-in IDE debugger dramatically decreases the performance. During the development of this project, the type of debugging had to be modified for the correct deployment of the application and this change resulted in slower response times from the application during debugging. Finally, when launching the universAAL server, we have had some difficulties when installing the service, since it requires a specific Java JDK version. **Connecting to the standard universAAL platform requires extensive knowledge and documentation about the API**.

Nowadays there is a plethora of IoT platforms which can benefit from the system proposed in this paper. For instance, the *Mozilla Webthings* (https://iot.mozilla.org, accessed on 1 November 2021) platform provides a service based on a gateway which can be installed in a RaspberryPi or a Turris Omnia router to control domotic assets connected to the internet using a wide range of wireless protocols such as ZigBee, ZWave and Wi-Fi. Another example is *Google Cloud IoT* (https://cloud.google.com/solutions/iot/, accessed on 1 November 2021) which provides a comprehensive set of tools for the integration, processing, storage and analytics of any type of data streamed from IoT devices. The system presented in this paper is a combination of these two paradigmatic solutions and customized the use of IoT technologies for the specific case of health and assisted living. Moreover, the integration with universAAL also attracts the interest of policy makers and responsible persons of a large deployment of AAL services that found in a universAAL open platform, interoperable with any prior or further investment in devices or services and free of vendor-lockings. For instance, a recent system constructed over universAAL described by Fortino et al. [33] sacrifices the interoperability at a sensor level and proposes the use of a closed set of health devices from specific vendors. This set-up, despite being promising, compromises the scalability in large scenarios which may require more flexibility in the sensor layer.

The application stores in the system the private information of doctors and patients, including phone numbers or emails. Privacy and security are two important factors to be analyzed in future works, and it is important to indicate what type of guidelines have been followed to increase the integrity of the system [34]. The data is stored in an SQLite database. This relational database manager has an RSA encryption algorithm. This is one of the most used encryption algorithms for digital signatures and is based on the factorization of integers. The deployment in UWP of the application also provides an apex of security to the system, since, in the case of wanting to take the data from it, a decompiler should be used to extract the information from the application. One of the reasons why we have chosen to deploy the system in a Raspberry Pi is because of its low energy consumption. A conventional computer consumes and average of 180–450 watts per hour while the Raspberry Pi 3B only consumes 4 W per hour [35], that is, 100 times less than a conventional computer.

### 5.1. Limitations

There are some limitations of the presented work which sould be taken into account, which are listed as follows. (1) The system has been designed as a conceptual architecture which integrates standards to be extended in the future. Therefore, this paper may be more useful for developers and designers, to whom this proposal may be useful to design interoperable eHealth systems, instead of end-users. (2) The system has been designed and implemented using Microsoft .NET technologies, using a local database (SQLite) for data persistence. The use of SQLite libraries has been chosen against a Relational Database Management System (RDBMS) such as MySQL due to the advantages it offers in its local use, focusing on read-write operations and increasing the portability of the system. (3) The system must communicate with the REST API of universAAL. Defining the communication of the system and how it interacts with these external systems is one of the major concerns of architecture. (4) We need to establish an ad hoc form of communication between the system and the sensors. Due to the behaviour of the system, it can be established that the sensors act as an active system, continuously sending information (PUSH); or by requesting the information when necessary from the system (PULL), that is, passively. (5) This paper does not consider the power consumption of the system, and this is a crucial aspect for its maintenance and validation in a real setting. Future works should study the optimal architecture configuration for keeping a balance between information flow and energy supply [16].

### 5.2. Future Work

The paradigm of the Internet of Things is constantly evolving and growing [36]. Recently, 5G technologies are being introduced in several use case scenarios, along with the release of the new Raspberry Pi lite versions. All these changes make scalability in major projects a very important point when creating a software system that supports this kind of technology [37]. Thus, the system was created in a way that could be extended in the future. Examples of this includes the following: (1) increase the number of IoT sensors that can be used to manage the application—either via the e-Health Kit or using a Bluetooth data protocol, the number of sensors allowed by the system can be increased without problems, since it has been designed to this end; (2) make a Web application—the data is transferred to the universAAL server, but it would be interesting to create a Web portal in which doctors could view the data from anywhere and at any time; and (3) export the application itself to other platforms and operating systems—an interesting idea would be to have not only the data received by universAAL, but to have an ASP.NET application, for example, in which they could host the application’s functionalities. In this way, the patient could choose between using the Raspberry Pi for monitoring, or in case of not being at home, to monitor their data remotely through any other computer.

## 6. Conclusions

This paper describes the design and implementation of an interoperable monitoring system between different IoT devices, integrating standards for the data exchange as FHIR and other systems for the Ambient Assisted Living such as universAAL. The system presented in this paper contributes to the widespread use of this standard for exchanging clinical information from home to the clinics, using low-cost IoT technologies. New e-Health systems for the remote monitoring of dependent patients and the elderly should consider the integration of extendable IoT technologies which can benefit the developers’ community, the creation of new open-specification hardware solutions and new tools to face the major challenges of an ageing society. The system includes two new features: (1) we propose a new layer based on rapid prototyping sensors to avoid limiting the data acquisition with specific vendors; and (2) the system is interoperable with FHIR standards and the universAAL platform, ensuring that adopters or developers of this system can build and adapt solutions to co-exist with existing systems/platforms/service.

## Figures and Tables

**Figure 1 sensors-22-01646-f001:**
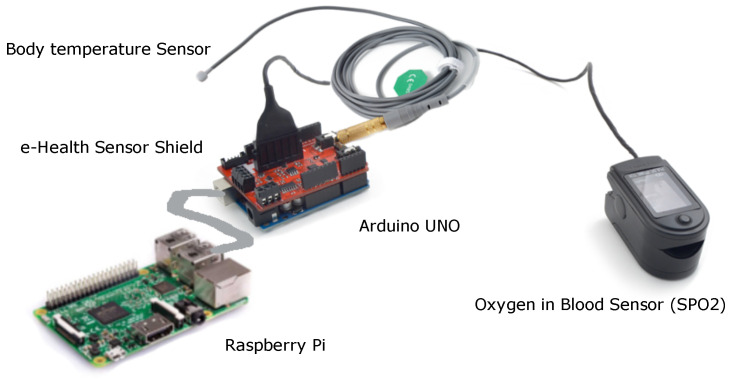
All the hardware components used in the system.

**Figure 2 sensors-22-01646-f002:**
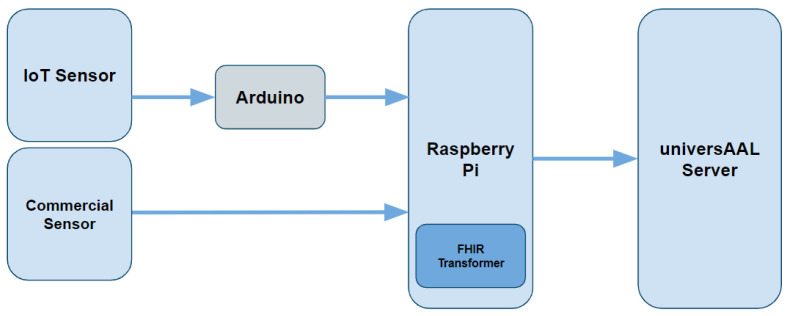
Architecture of the IoT-Health care system.

**Figure 3 sensors-22-01646-f003:**
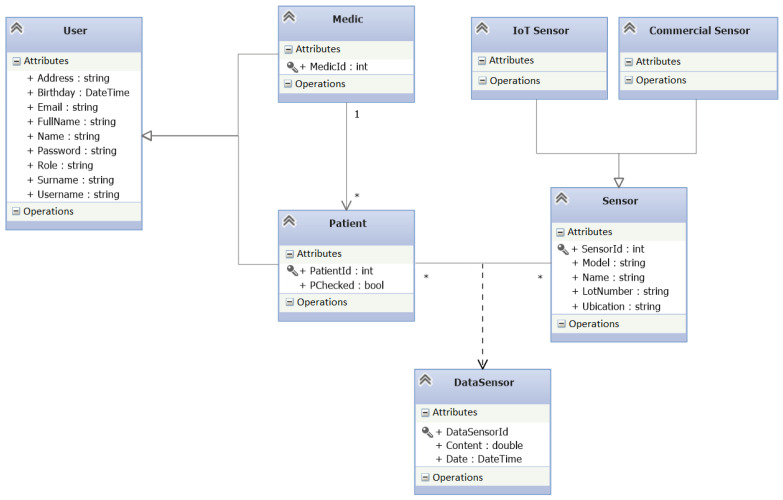
Database diagram of the system.

**Figure 4 sensors-22-01646-f004:**
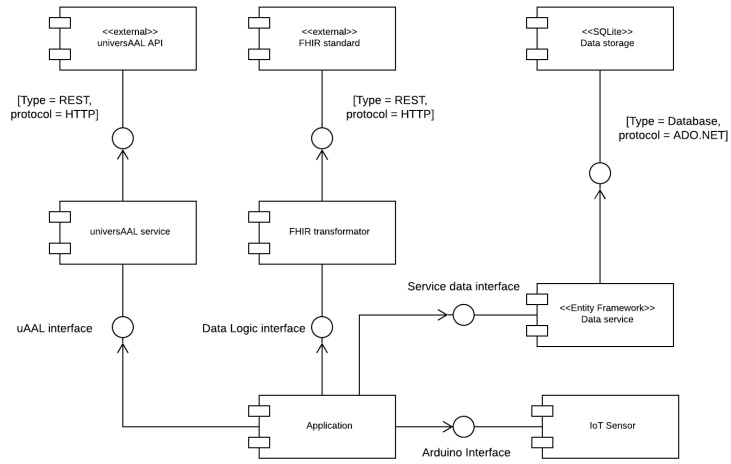
Logic diagram of the system components.

**Figure 5 sensors-22-01646-f005:**
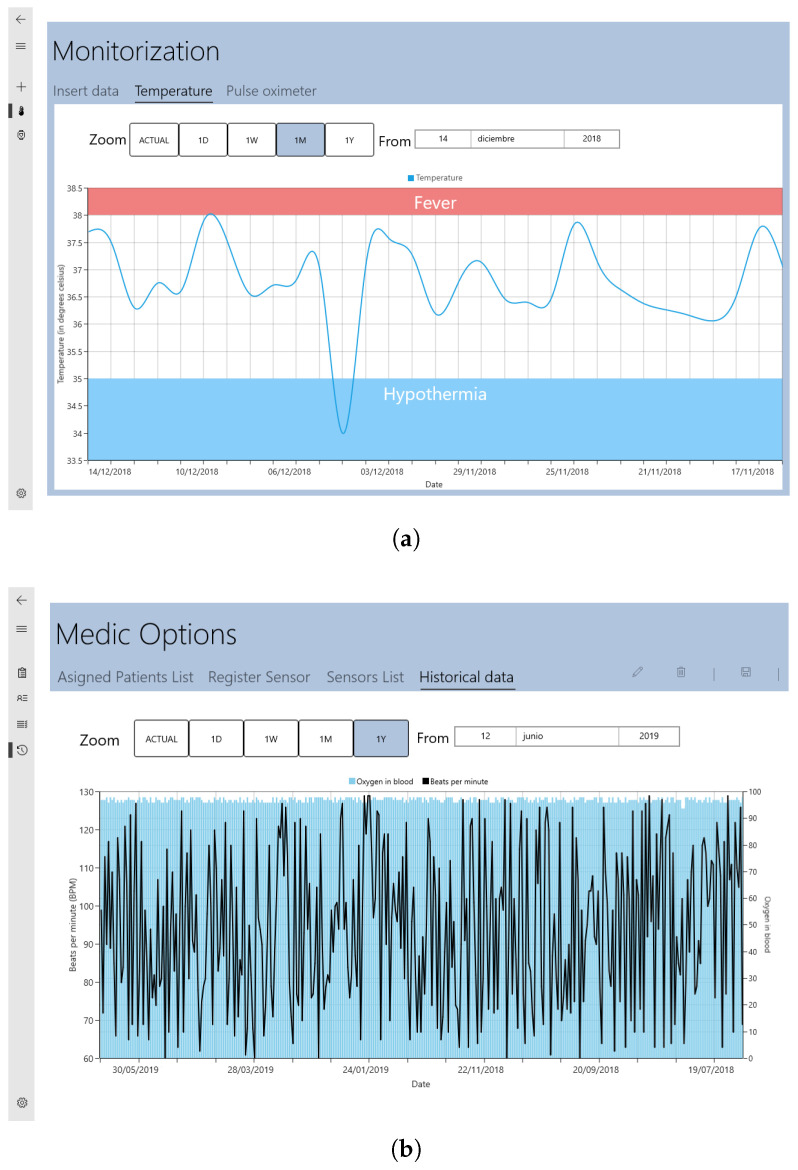
Visualisation of historical data by role (**a**) patient, (**b**) doctor.

**Figure 6 sensors-22-01646-f006:**
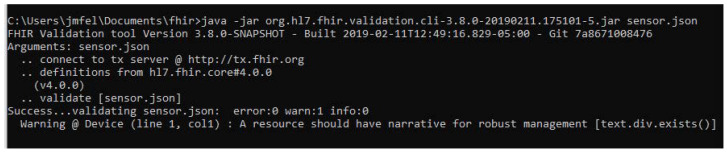
The FHIR validator consists of a “.jar” file which validates the resources that are in JSON format.

**Figure 7 sensors-22-01646-f007:**
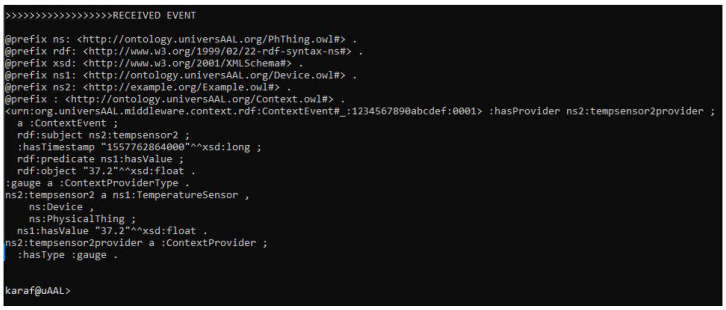
Event of data registration in universAAL.

**Table 1 sensors-22-01646-t001:** Comparison among the state of the art existing solutions focused on IoT and Health.

Study	Standard	Scalability	Persistence	Portability	Efficiency
[3]	None	High	High	Platform	Not reported
[11] FHIR	None	MongoDB	None	High	
[12]	FHIR	Low	Low	Raspberry Pi 2	High
[13]	FHIR	Medium	High (SQLite)	Smartphone	Depends on the device
[14]	FHIR	High	Low	Smartphone	Depends on the device

**Table 2 sensors-22-01646-t002:** Use cases considered for the definition of the system.

Use Case	Description
User authentication	Once the user is registered in the application, the credentials are verified and the user is redirected to the main page of the application. **Pre-condition**: a user must be registered by the system administrator
Data acquisition	Once the patient has logged in, they can access the monitoring functions of the application. The health professional may also access the monitoring functions of his assigned patients. To do this, an historical data tab should provide navigation functions to select a patient and a sensor. The patient can also enter data manually into the application. **Pre-condition**: successful log in
Sensors management	The health professional is in charge of the management of sensors in the application, therefore, once logged in, sensors assigned to patients can be registered, modified or removed. To modify or remove a sensor, the user will have to select it from the sensor list. **Pre-condition**: authentication as health professional
User management	The system administrator is responsible for registering, unregistering and modifying the users of the application. If a user is a patient type, they must have an assigned doctor; if the user is a doctor, it can contain a list of assigned patients. To modify a user, the administrator will have to select it from the list of users.
Interoperability	Each data point acquired by the sensor and monitored by the application is stored in the local database. Once the data have been validated by the FHIR standard, a request to the universAAL REST API is made using a POST method, which will vary depending on the type of sensor. On the server, the system must have a Publisher that will send all the data to all Subscribers who are subscribed to the universAAL service. Any external agent that meets the requirements as a Subscriber may use the service and deploy it to another system.

**Table 3 sensors-22-01646-t003:** Summary of the main components used in the system.

Component	Functionality
UniversAAL API	Provides the necessary methods for client–server communication between the application and the universAAL service. Handles all communications through a REST API.
UniversAAL service	Manages the sending of data between the UniversAAL server and the application through the use of POST, UPDATE and DELETE methods.
FHIR Standard	Provides a standard for the exchange of patient data.
FHIR Transformer	Makes use of the standard libraries to transform the data. It is responsible for the correct verification of the data before it can be sent to the server.
Data Storage	Provide persistence to the application thanks to the use of a local database in SQLite.
Data Service	Provides the necessary libraries for the use of methods related to the persistence of data with the Entity Framework.
IoT sensor	It is responsible for collecting the data of each patient.

## Data Availability

Not applicable.

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
