# Peer review of "A Proof-of-Concept IoT System for Remote Healthcare Based on Interoperability Standards"

_sensors, 2022, doi:10.3390/s22041646_

Round 1

Reviewer 1 Report

The scope of the paper is to design and deploy an interoperable system based on IoT for the telemonitoring of elderly people in home environments. The idea is not new, there are many papers dealing with the same topic, so the state of the art should be more elaborated.

The third objective you declared, respectively: "To guarantee the interoperability of the system through the implementation of standards capable of allowing the integration of the project with other systems, based on the same standard." it's trivial, as the scope of a standard is to provide a framework for all devices operating following the respective standard to be compatible one with each other. There is nothing to propose here, nothing to demonstrate, simply follow the standards' requirements and that's all.

Regarding the state of the art: You should revise this section as there are major drawbacks. Searching for scientific papers does not mean you cover the technologies in a field, for the single reason most of the equipment and devices are using proprietary standards that are not disclosed to the public and are not presented in the scientific literature. There is a major gap justified by the commercial interest of the actors in the field. 

The XIAOMI MyFit application connects by using Bluetooth to Xiaomi proprietary devices, and these devices are not IoT ones as they do not connect to the Internet by themselves. XIAOMI produces IoT devices (maps, vacuum cleaners. etc.) but all health devices may not be characterized as IoT.

For the proposed framewrok, 4 roles are assigned for every category of rights, one role is for interoperability. 

The hardware components presented in a paper is around a Raspberry Pi microcomputer, connected with an Arduino Uno, which connect all the sensor. As the Raspberry has sufficient resources to accomodate many sensors sending a large variety of signal, one may consider the use Arduino not justified. It is good to justify at large the topology of the hardware and the role of each components.

There is no information on the power supply used for the RP and the sensors, this being a key issue for every IoT application. The rest of the paper concentrated on the software component. 

There are many affirmations that are simply put in the text, without any explanations or demontrations they are valid. Some of them are tehnically wrong. Several examples (despite the bad English):

(322) "We have been able to observe that these response times improve greatly when deploying the application on the Raspberry Pi"

(338) "The system was designed as a concept of a project that can be extended in the future, that is, with high scalability"

(360) "In front of a conventional computer, which consumes 1.1kW every hour"

I don't understand what is this phrase for and how the reader may use this piece of information:

(324) "It also requires knowledge about networks and ports, which the student obtained during the deployment of the service."

The paper needs a serious English language revision.

Author Response

Reviewer #1

The scope of the paper is to design and deploy an interoperable system based on IoT for the telemonitoring of elderly people in home environments. The idea is not new, there are many papers dealing with the same topic, so the state of the art should be more elaborated.

Thank you very much for this comment. We have reviewed the introduction and the motivation of the work to stake the two differential points of the presented system: to implement an interoperable healthcare information exchange protocol (FHIR) and to be integrated with a reference architecture for Ambient Assisted Living, such as UniversAAL.

The third objective you declared, respectively: "To guarantee the interoperability of the system through the implementation of standards capable of allowing the integration of the project with other systems, based on the same standard." it's trivial, as the scope of a standard is to provide a framework for all devices operating following the respective standard to be compatible one with each other. There is nothing to propose here, nothing to demonstrate, simply follow the standards' requirements and that's all.

We agree with the reviewer’s comment and we have erased this naïve objective.

Regarding the state of the art: You should revise this section as there are major drawbacks. Searching for scientific papers does not mean you cover the technologies in a field, for the single reason most of the equipment and devices are using proprietary standards that are not disclosed to the public and are not presented in the scientific literature. There is a major gap justified by the commercial interest of the actors in the field.

The section has been reviewed and improved

The XIAOMI MyFit application connects by using Bluetooth to Xiaomi proprietary devices, and these devices are not IoT ones as they do not connect to the Internet by themselves. XIAOMI produces IoT devices (maps, vacuum cleaners. etc.) but all health devices may not be characterized as IoT.

We agree on this comment and we have removed XIAOMI device because it is not connected to the Internet, and thus, cannot be considered as IoT.

For the proposed framework, 4 roles are assigned for every category of rights, one role is for interoperability. The hardware components presented in a paper is around a Raspberry Pi microcomputer, connected with an Arduino Uno, which connect all the sensor. As the Raspberry has sufficient resources to accomodate many sensors sending a large variety of signal, one may consider the use Arduino not justified. It is good to justify at large the topology of the hardware and the role of each components.

This is a good point and actually was an issue we have discussed largely. We came to decide that sensors must be equipped with a sort of computational and connection capability, so they can process information ( send only meaningful information to the Raspberry) and also connect through wireless channels, so the sensor can be in another location different from the central station (Raspi). We have added a new paragraph in the discussion elaborating on this.

There is no information on the power supply used for the RP and the sensors, this being a key issue for every IoT application. The rest of the paper concentrated on the software component.

In this paper we have focused on the concept of the architectures and the software characteristics for making it interoperable. A new limitation has been added in the discussion to identify the needs of studying the energy consumption, specially in an IoT environment. Thank you for the remark.

There are many affirmations that are simply put in the text, without any explanations or demontrations they are valid. Some of them are tehnically wrong. Several examples (despite the bad English):

(322) "We have been able to observe that these response times improve greatly when deploying the application on the Raspberry Pi"

This sentence is to confirm that response time decreased significantly when moving from a testing environment (debugging) to a deployment setting (release). Anyway we have removed this misleading sentence.

(338) "The system was designed as a concept of a project that can be extended in the future, that is, with high scalability"

In this sentence we wanted to confirm that the purpose of the architecture and hardware is to be efficient (low cost). The scalability is ensured by using a distributed agent-based architecture and cheap hardware (Sensors and computing units). We have eliminated the word scalability to focus on the limitation, that is, we want future developers to build on top of this concept.

(360) “In front of a conventional computer, which consumes 1.1kW every hour”

Some of the systems we have studied in the state of the art use a personal computer as middleware to concentrate the devices. In this comparison we want to highlight the gains of using a microPC as the raspberry pi as al alternative. We have rephrased the sentence.

I don't understand what is this phrase for and how the reader may use this piece of information:

(324) "It also requires knowledge about networks and ports, which the student obtained during the deployment of the service."

We wanted to state the difficulties of connecting to UniversAAL ( a platform endorsed by the European Commission for research initiatives in the scope of Ambient Assisted Living). We have written again the sentence.

The paper needs a serious English language revision.

Thank you very much, the writing has been revised in deep.

Reviewer 2 Report

Authors have highlighted the emerging and core issue, but still there are major issues to be fixed.

Reviews to Authors

  • Title must be simple, clearer and nicer.
  • Spell out each acronym the first time used in the body of the paper. Spell out acronyms in the Abstract by extending it.
  • The abstract can be rewritten to be more meaningful. The authors should add more details about their final results in the abstract. Abstract should clarify what is exactly proposed (the technical contribution) and how the proposed approach is validated.
  • What is the motivation of the proposed work?
  • Introduction needs to explain the main contributions of the work clearer.
  • The novelty of this paper is not clear. The difference between present work and previous Works should be highlighted.
  • Authors must explain in detail the introduction section.
  • Authors must develop the framework/architecture of the proposed methods
  • There is need of flowchart and pseudocode of the proposed techniques
  • Proposed methods should be compared with the state-of-the-art existing techniques
  • Research gaps, objectives of the proposed work should be clearly justified.

To improve the Related Work and Introduction sections authors are highly recommended to consider these high quality research works <A Novel Adaptive Battery-Aware Algorithm for Data Transmission in IoT-Based Healthcare Applications, Electronics, MDPI, Vol.10, No.4, pp.367, 2021 >, <An Energy-Efficient Algorithm for Wearable Electrocardiogram Signal Processing in Ubiquitous  Healthcare Applications”, MDPI Sensors Vol.8, No.3, pp.923, 2018 >

  • English must be revised throughout the manuscript.
  • Limitations and Highlights of the proposed methods must be addressed properly
  • Experimental results are not convincing, so authors must give more results to justify their proposal.

Finally, paper needs major improvements

Author Response

Reviewer #2

Title must be simple, clearer and nicer.

Thank you for taking the time of reviewing our submission, we feel that these comments have helped to improve the manuscript and to make clear the presentation of our work. We would like to continue improving the paper, so more advice on the title changes would be much appreciated.

Spell out each acronym the first time used in the body of the paper. Spell out acronyms in the Abstract by extending it.

The abstract can be rewritten to be more meaningful. The authors should add more details about their final results in the abstract. Abstract should clarify what is exactly proposed (the technical contribution) and how the proposed approach is validated.

We have revised and written again the abstract

We have extended the acronyms in the abstract and revised it to include the proposed information

What is the motivation of the proposed work? Introduction needs to explain the main contributions of the work clearer. The novelty of this paper is not clear. The difference between present work and previous Works should be highlighted. Authors must explain in detail the introduction section.

The introduction has been re-structured and revised in deep. Now the discussion contains a comparison of the previous work and the differential points of the proposed system

Authors must develop the framework/architecture of the proposed methods. There is need of flowchart and pseudocode of the proposed techniques Proposed methods should be compared with the state-of-the-art existing techniques

We agree with the reviewer that a high level architecture could be of interest to explain the whole system, however we believe that Figures 1 and 2 are enough to show the simplicity of the system, and such an schema would be very technical (as another reviewer says).

Research gaps, objectives of the proposed work should be clearly justified.

The discussion has been enhanced to consider these crucial points, thank you very much for the comment.

To improve the Related Work and Introduction sections authors are highly recommended to consider these high quality research works <A Novel Adaptive Battery-Aware Algorithm for Data Transmission in IoT-Based Healthcare Applications, Electronics, MDPI, Vol.10, No.4, pp.367, 2021 >, <An Energy-Efficient Algorithm for Wearable Electrocardiogram Signal Processing in Ubiquitous  Healthcare Applications”, MDPI Sensors Vol.8, No.3, pp.923, 2018 >

We would like to thank the reviewer for these nice sugestions, we have analyzed them and added as references in our paper, specially the paper related to an efficient algorithm for power-supply management.

English must be revised throughout the manuscript.

The writing has been revised in deep.

Limitations and Highlights of the proposed methods must be addressed properly

We have revised this paragraph in deep.

Experimental results are not convincing, so authors must give more results to justify their proposal.

We agree that our laboratory validation may seem limited. However we wanted to publish this proof of concept to afterwards perform a real validation with users in a pilot test.

Finally, paper needs major improvements

Reviewer 3 Report

This paper presents the design and implementation of a health monitoring system based on the integration of rapid prototyping hardware and interoperable software to build and test a system with several sensors for the remote transmission of environmental and physiological variables.

The work is very interesting but it lacks European efforts in the area like the  recent work of 7 European projects in the area of Healthcare (electronics10141616) another effort could be in the area of SDN in healthcare as an enabler for connecting IoT devices. An intersting work can be An NFV-powered emergency system for smart enhanced living environments where the Network function virtualisation is used as an enabler in IoT monitoring of the healthcare infrastructure.

Next the paper is missing a high level architecture showing the interconnection of the IoT devices with the core system and the figure 1 is too technical for a journal publication.

Figure 3 does not provide anything to the reader of the paper.

Figure 6 b Is not showing anything and is too dense to follow.

Author Response

Reviewer #3

This paper presents the design and implementation of a health monitoring system based on the integration of rapid prototyping hardware and interoperable software to build and test a system with several sensors for the remote transmission of environmental and physiological variables.

The work is very interesting but it lacks European efforts in the area like the  recent work of 7 European projects in the area of Healthcare (electronics10141616) another effort could be in the area of SDN in healthcare as an enabler for connecting IoT devices. An intersting work can be An NFV-powered emergency system for smart enhanced living environments where the Network function virtualisation is used as an enabler in IoT monitoring of the healthcare infrastructure.

Thank you very much for the comments and the literature suggestions. We have reviewed them, especially the overview of platforms and architectures and added it to the introduction and the discussion of the paper.

Next the paper is missing a high level architecture showing the interconnection of the IoT devices with the core system and the figure 1 is too technical for a journal publication.

We do not agree with this comment, as this figure shows the simplistic set up of the proposed system, which will be in the homes of the dependent patients.

Figure 3 does not provide anything to the reader of the paper.

We agree with this comment and we have removed the picture

Figure 6 b Is not showing anything and is too dense to follow.

In this figure we wanted to show the prototype of the GUI, as the text says it is just a representation of the patient and doctor views ( no needs to interpret the data within).

Thank you very much for the review and the positive literature suggestions.

Round 2

Reviewer 1 Report

Dear Authors,

By examining the responses, I would make several comments on the modifications only, which are supposed to make the paper better.

1. You maintained the affirmation (328): "It also requires knowledge about networks and ports, which the student obtained during the deployment of the service." I made an observation regarding the word "student" who obtained knowledge about networks and ports.  For the reader, there are only the authors of the paper, they are not aware of any student involved.

2. You also maintained the affirmation (363): "In front of a conventional computer, which consumes 1.1kW every hour ..." - You have to note a conventional computer DOES NOT consume 1,1kWh. A standard power supply is only about 250W, for a more complex configuration 450W is enough. I've mentioned in my previous comments, but you choose to preserve this. This is presented in the conclusion section as a major advantage of your solution.

3. You still mentioned nothing about how the device is powered. You chose a configuration with two controllers, Arduino and Raspberry Pi, and the energy requirements of both (including the sensors) are pretty high, and the ensemble is hard to be viewed as an IoT device, which is supposed to operate for several years on battery - hard to believe, with about 5W consumed. After considering this, all the rationale behind the paper is very hard to accept.

Moreover, there is no real comparison with other hardware equipments, already used for these kind of tasks. You refer in Table 1 several systems, but the real advantages of your solution when compared with others are not presented in the paper, on a point by point basis (what did other wrong and what is better in your solution). Also, in Table 1 you still mention the Xiaomi Myfit, which we both agreed it is not an IoT system (see your answer to my previous review).

By reading the paper, one may understand you propose a combined hardware & software solution to monitor two body sensors (temperature and pulseoxymeter). In my opinion, there is no need to have this mix, you better concentrate on the software part, considering only the information you receive from a piece of hardware, without entering into details. This is not diminishing the contributions of the paper as you may concentrate to explain at large the originality of the software you propose, and compare with other in terms of the software.

Overall, the paper needs real improvements in order to be accepted.

English language should be revised.

Author Response

Please see the attachment and please note that the latest version incorporates all the suggested changes.

Reviewer 2 Report

Paper lacks the novelty, significance and strong motivation

Results section is very weak, so authors are hihgly recommended to add the results (whether hardware or software)

Introduction section is shorter, so must be extened 

Related work section is missing, so must be created separtely

Comparision with the state of the art exisitng works must be presented for the justification of their proposal and its performance

To strengthen the Intro and related work section authors are highly suggested to add these high quality works <,  Dynamic Application Partitioning and TaskScheduling Secure Schemes for Bio-Sensors Healthcare Workload in Mobile Edge Cloud, Electronics, 2021>, <AI-Enabled Framework for Fog Computing Driven E-Healthcare Applications>

Paper cannot be accepted in its present form

Reviewer 3 Report

Although the authors proceed with most of the comments the suggestions to add literature from the European efforts like e.g.

" Reference Architectures, Platforms, and Pilots for European Smart and Healthy Living—Analysis and Comparison" doi :10.3390/electronics10141616

or other work that  is currently is supported in US like:

Petersen, A., Davis, M., Fraser, S., & Lindsay, J. (2010). Healthy living and citizenship: An overview. Critical Public Health, 20(4), 391-400.

This small addition can be added in the finalisation phase of the paper.
